# Effect of Ginseng Intake on Muscle Damage Induced by Exercise in Healthy Adults

**DOI:** 10.3390/nu16010090

**Published:** 2023-12-27

**Authors:** Borja Muñoz-Castellanos, Patricia Martínez-López, Rafael Bailón-Moreno, Laura Esquius

**Affiliations:** 1Faculty of Health Sciences, Universitat Oberta de Catalunya, 08018 Barcelona, Spain; borjamc25@gmail.com (B.M.-C.); lesquius@uoc.edu (L.E.); 2“Techné” Knowledge and Product Engineering Research Group, Faculty of Science, Universidad de Granada, 18071 Granada, Spain; bailonm@ugr.es

**Keywords:** ginseng, muscle damage, supplements, exercise nutrition

## Abstract

One of the most popular herbal supplements in the world is ginseng. Several studies have evaluated the capacity of ginseng as a protective element in the physiological response to exercise. The result produced by the exercise causes an increase in cellular biomarkers of damage in the skeletal muscle, mainly in the pro-inflammatory types. The different types of ginseng are composed of ginsenosides, which are active ingredients that act on the central nervous system and have antioxidant and anti-inflammatory properties, as well as effects on cortisol modulation. The use of ginseng as a nutritional supplement can help muscle regeneration and renewal. The objective of this review is to enrich the knowledge regarding the consumption of ginseng for a specific situation, such as exercise, which would cause an improvement in the tolerance to chronic load stimuli in sport, thus helping the subjects to recover between training sessions. Due to these benefits, it could also be an ideal food supplement for regenerative processes in muscle injuries in which inflammatory markers increase significantly. This review aims to summarise that biological factors can be attenuated after exercise due to the consumption of ginseng in healthy subjects, accelerating and improving muscle regeneration and, therefore, improving the ability to adapt to the stimuli generated by said exercise.

## 1. Introduction

### 1.1. Ginseng—Herbal Supplement

One of the most popular herbal supplements in the world is ginseng [1]. The different species of ginseng are specific to the place of production, providing different benefits depending on the species [2]. Particularly in Asian countries, such as China and Korea, it is used in medicine and nutrition due to its great and diverse benefits [3]. These benefits are determined by the variety of nutrients that make it up, such as vitamins, minerals, fiber, proteins, saponins and ginsenoside [3]. Its main active substance is ginsenosides, of which, to date, 112 types have been discovered, isolated, synthesised and metabolised [4]. The most studied have been the ginsenosides Rb1, Rg1 and Rg3 [4,5]. Their benefits are identified as anti-inflammatory, antioxidant, brain function stimulant, anabolic and immunostimulant and, because of all this, better physical performance [3].

Various studies have evaluated the capacity of ginseng as a protective element in the physiological response to exercise [6]. Additionally, ginsenosides can improve mitochondrial function, promoting protein synthesis, increasing myoblast differentiation and inhibiting protein degradation [6].

### 1.2. Exercise and Its Metabolic Processes

All muscular actions—concentric, eccentric and isometric—have the capacity to generate muscle damage [7]. The result produced by exercise causes an increase in cellular biomarkers of damage in skeletal muscle, mainly in pro-inflammatory ones [8]. The inflammatory response process due to exercise is a necessary element for triggering the process of muscle repair, regeneration and adaptation [9]. This muscular adaptation generates an increase in the cross-section of the muscle called hypertrophy and, consequently, an increase in physical performance [10]. The damage generated is specific to each macromolecule of the tissue and induces injuries to the contractile elements and the cytoskeleton. The specific isoform of the cytoskeleton is creatine kinase (CK). Its three types are located at different points of the muscle fibers. The different CK isoforms provide specific information on the localisation of stress induced in the tissue [7] and depend on various factors that individually affect each subject and are specific to exercise [7]. Once this muscle damage has occurred, the level of CK in the body increases in the hours after exercise, and its peak is generated between 24 and 72 h afterwards [11]. Despite the damage caused, muscle is a tissue with a high capacity for adaptability to the stress produced by contraction, which provokes an anabolic response in the body to improve stressed structures [12]. This metabolic process decreases the contraction capacity of muscle fibers and is called muscle fatigue, which causes a feeling of tiredness and lack of energy in those who experience it. The main causes of this state of fatigue are related to overtraining, detraining or injuries [13].

### 1.3. Ginseng and Exercise

Traditional Chinese Medicine and herbal medicine philosophy consider ginseng to be an adaptogen that helps restore balance in the body [14]. Ginseng is used as an herbal supplement that helps increase energy—both physical and emotional—and general well-being [15]. The different types of ginseng are composed of ginsenosides, active agents that act on the central nervous system and have antioxidant and anti-inflammatory properties, as well as effects on cortisol modulation [14]. The use of ginseng as a nutritional supplement can help muscle regeneration and renewal [6]. There is sufficient evidence on the biological stress generated by exercise, which immediately causes a catabolic process in the organism necessary to be able to adapt to it and, consequently, generate development of the structures involved [7]. The objective of this review is to enrich the knowledge regarding the consumption of ginseng for a specific situation, such as exercise, which would cause an improvement in the tolerance to chronic load stimuli in sport, thus helping the subjects to recover between training sessions. Due to its aforementioned benefits, it could also be an ideal food supplement for regenerative processes in muscle injuries in which inflammatory markers increase significantly [16].

This review aims to summarise that biological factors can be attenuated after exercise due to the consumption of ginseng in healthy subjects, accelerating and improving muscle regeneration and, therefore, improving the ability to adapt to the stimuli generated by said exercise.

## 2. Materials and Methods

This review summarises the studies found to date in which the use of ginseng as a nutritional supplement is used to verify the effects on the physiological response to exercise.

### 2.1. Goals

The main objective of this review was to identify whether there is scientific evidence that justifies the regenerative capacity of ginseng in the process of muscle damage induced by exercise. To support this main objective, specific objectives were established, such as:-To evaluate the benefits of ginseng in the body after exercising.-To evaluate the capacity of ginseng intake on the regeneration of muscle damage induced by exercise.-To evaluate the suitability of ginseng intake in the process of recovery from muscle injury.

### 2.2. Researchable Questions

Using the PICO question structure as a reference, the following questions were raised in this review:-What physiological benefits can healthy adults obtain from the regular intake of ginseng compared to those who do not ingest this supplement?-Can the muscle damage induced by training be attenuated by the consumption of ginseng compared to its non-intake in healthy adults?-Is there a positive relationship between ginseng intake in the recovery of muscle injuries in healthy adults compared to those who do not consume it?

### 2.3. Search Strategy

In the first place, for the compilation of the scientific documents necessary for this review, the database was determined due to the characteristics of the subject exposed, SCOPUS. The database chosen for the preparation of this systematic review provided a very complete source of articles in which all types of articles appear and from which others, such as PubMed and ScienceDirect, directly related to the processes studied in this review, are fed. The search deadline was set as April 2023, with no deadline regarding its age. The main search algorithm used was “ginseng AND sport AND muscle damage”. The reference list of the chosen articles was also taken into account. Only articles whose language was English or Spanish were selected.

### 2.4. Selection of Studies

There were 766 identified studies. Of all of them, 12 were included in this review (Figure 1, PRISMA flowchart). Next, the full text of all articles obtained as a result of said screening was reviewed. Once obtained, they were critically read and the data were extracted, ensuring in this last phase the suitability of the inclusion criteria with the selected articles.

All 12 articles were described by the authors as “randomised” and “double blind”; 7 of them were labeled as “crossover design”; 4 were described as “parallel design”; and only 1 was labeled as “counterbalanced design”.

Of all items included, 8 of them were carried out in Asia, 2 in the USA, 1 in Australia and 1 in Brazil. The studies were published from 2005 to 2021.

After the search was carried out in the database, the articles were screened according to their title and “Abstract”. This first screening was carried out according to the eligibility criteria shown in Table 1. This table provided a response to the target population to which the review process is directed. Once the population was chosen, the exposure factor was defined, which was composed of the consumption of ginseng as a dietary supplement. The comparison between groups that consumed or did not consume said supplement defined the structure of the review in this aspect. The main focus of the results obtained was directed towards the physiological processes generated by the intervention of ginseng consumption after induced exercise. The choice of the types of trials selected was shown in a generalised manner, being mainly randomised controlled trials and cohorts with experimental and control groups. For this screening, the Beta version program Abstrakr was used. This selection was made by only 1 reviewer, based on the inclusion and exclusion criteria of articles shown in Table 2. In this form, a description of the questions that define the research of this review is provided. Subsequently, the necessary information was developed to consider each of the articles as included or excluded articles. Next, the design of the selected articles was subjected to review, considering only those that were carried out with humans and came from a systematic review or meta-analysis. Observational studies were chosen only if they had an experimental group and a control group and cohorts. Regarding the design of the study, it must be randomized and, at least, blind, including those that are double-blind. Finally, possible cases of doubt are explained, including how they were resolved for their definitive inclusion or exclusion from the review. To make an objective selection, an article selection form was structured.

### 2.5. Data Extraction

To identify the key elements from existing guidelines and texts, as well as from relevant systematic reviews, we followed the same data extraction procedure as in a previous systematic review, Data extraction form—Scoping ReviewWG3-A8 [17]. The aim of this scoping review is to assess the resource intensity of each step of the systematic review process. In addition, this protocol focuses on reasons why diverse steps of the systematic review process are resource-intense. This data extraction form was used, reviewed and refined by the authors to better capture the key aspects that are essential for evaluation, synthesis and presentation, ensuring the adequacy of the tool.

In this protocol, the aim of this study was established. Subsequently, the area corresponding to the revised studies was defined. Then, the systematic analysis was focused on intervention articles. Identifying the number of studies used and, if any, the number of systematic reviews. Finally, the steps taken in the review process were evaluated. Steps from this protocol have been represented in Table 3.

### 2.6. Data Collection

We extracted the following data from included studies using a standard form:Author, year and countryStudy designCondition treated (type, doses and duration)Compared groupsPerformance testOutcome measuresAuthors’ conclusions

### 2.7. Quality Assessment

To evaluate the methodological quality of the included studies, the Cochrane Collaboration Risk of Bias Tool (CCRBT) for RCTs was used [18]. The Cochrane Collaboration strongly encourages all reviewers to use these tools to establish consistency and avoid discrepancies in the assessment of methodological quality among all review groups. The methodological quality of each study was independently evaluated by two review authors (BMC, PML) using the assessment tools. Disagreements were resolved by discussion until a consensus was reached. A senior expert (LE) contributed to the assessment procedure whenever it was considered necessary. The result of the quality assessment from this tool is presented in Figure 2. 

### 2.8. Participants

A total of 276 subjects, including men and women, were involved in these studies. The sample size varied between 11 and 21, with only the exception of one article that included 110 subjects.

All studies were carried out with healthy subjects, of which 4 of them were described as physically active, and 2 of the studies reviewed were carried out with athletic subjects.

### 2.9. Intervention

All articles reviewed used ginseng ingestion in any variety. Eight different types of preparation or extracts were used in these studies. Korean Ginseng (*n* = 3), American Ginseng (*n* = 2), Red Ginseng (*n* = 2), Panax Ginseng (*n* = 1), Panax notoGinseng (*n* = 1), Wild Ginseng Extract (*n* = 1), Jiling Ginseng (*n* = 1) and Rb1 (*n* = 1). All ginseng supplements were ingested orally.

## 3. Results

The results of this bibliographic review show the data obtained from the available information on the relationship between physical exercise and ginseng consumption. These data are based on studies in which ginseng is ingested together with a physical exercise intervention to observe the physiological response caused by this herbal supplement. The results of the effects of ginseng consumption are largely beneficial for a reduced appearance of fatigue and, consequently, better muscle recovery.

The systematic reading of the articles selected for this review provided an exhaustive analysis of each of them. This analysis can be seen in Table 4. In general, there are positive results regarding the influence on the attenuation of muscle damage, as well as an improvement in the subsequent regeneration of the organism and the subjective perception of effort. Consequently, physical performance can be improved, and there is also a decrease in the risk of injury. All these results are mainly induced by the lower appearance of fatigue after exercise. In a more in-depth way, the systematised study of the research is described below.

In 2005, Cheng-Chen Hsu et al. [19] published the first article evaluating the intake of ginseng in humans and its effect on the body after physical exercise. This article was carried out with 13 active male subjects in which a randomised, double-blind and crossover design of the groups was used. The ginseng variant used in this study was American Ginseng (AG), in which an intake of 1.6 g/day was carried out for 4 weeks, with a week of cleansing between intake cycles. Subsequently, an incremental treadmill test was performed at 80% of VO2max. Evaluation of the results obtained from the blood test indicated a lower increase in CK in the experimental group. The authors of this study concluded that the use of GA intake attenuates muscle damage induced by physical exercise.

Hyun Lyung Junget et al. [20], in 2011, carried out a randomised, double-blind study in Korea with 18 healthy male subjects where the intake was evaluated; on this occasion, it was red ginseng (RG) in daily doses of 20 g for 7 days. The physical activity that induced the physiological response was two sets of 45 min of running on an ascending treadmill (15% gradient at 10 km/h) with 5 min of rest between sets. The physiological data obtained were extracted through blood analysis, focusing attention on muscle damage, inflammation and fatigue, in addition to glucose and insulin levels. The results obtained from this study reflected an improvement in CK levels in the experimental group compared to the control group at 72 h. Regarding IL-6, the results defined an improvement in the levels of the experimental group compared to the control group at 2 h and 3 h after exercise. On the other hand, the experimental group obtained a lower insulin level after 90 min of exercise and a reduction in glucose levels after 60 min of exercise compared to the control group. In conclusion, these authors highlighted that ginseng intake attenuates muscle damage and inflammation induced by physical exercise.

Kate L. Pumpa et al. [21], in 2013, carried out the first study with a double-blind intervention design and with parallel groups of Australian athletes to evaluate the effect of ginseng intake on exercise-induced muscle damage. In this study, 20 male subjects were randomly distributed into two groups, control and experimental. The experimental group consumed four capsules of 1 g of Panax notoginseng 1 h before physical exercise, another four capsules immediately after and four capsules every 4 h until 48 h after exercise. The exercise test consisted of five series of 8 min running with a downward inclination (−10%) with 80% of HRmax. In this study, the authors evaluated, in addition to the blood analysis as in previous studies, the subjective perception of effort after performing the exercise and performance measures through performing jumps (CMJ and SJ). The conclusions of this intervention did not find significant data on ginseng intake in trained athletes.

In 2016, Seongeon KIM et al. [22] re-evaluated the action of RG in healthy subjects. In this randomised, double-blind study with a crossover protocol, 11 men were used in an intervention with 5 g of RG prior to performing a cycle ergometer test (Wingate Test) in which peak power and average anaerobic power were evaluated through the analysis of VO2max. Blood samples were taken after exercise at 30 min and 60 min, where the levels of lactate, ammonium and muscle damage (CK and lactate dehydrogenase) were analyzed. These authors found no differences in markers of muscle damage. On the other hand, RG intake decreased ammonium and lactate levels at 30 min and at 30 and 60 min, respectively. In conclusion, these authors reflect an improvement in the physiological markers related to muscle fatigue.

Up to this point, all the articles reviewed compared a given dose of ginseng supplementation with a control group. Shawn D. Flanagan et al. [23] went a step further in specificity by comparing 19 subjects, 10 women and 9 active men, divided into groups with two different doses of KG and a control group. The three groups were divided according to the intake consumed; those who took high doses (HD) took 960 mg/day, while the low-dose group (LD) had an intake of 160 mg/day, and finally, the third group was administered a placebo. The exercise chosen to induce physiological stress was strength training consisting of five series of 12 repetitions of leg press at 70% RM. Blood measurements were taken before resistance exercise, immediately after, 30 and 60 min after, and 24 h after. Questionnaires were also carried out to assess subjective measures such as mood and sleep quality. The results of this study show that HD intake attenuated the cortisol and CK response to exercise. In the same way, the group that consumed HD of KG increased glutathione in the 30 and 60 min after the exercise. Finally, HD increased the response of SOD (superoxide dismutase) and TAP (total antioxidant potency in plasma). They did not find changes in the emotional state or in the quality of sleep due to the intake of KG. The authors concluded that the ingestion of high doses of KG decreases muscle catabolism induced by exercise in the same way that it improves the acute antioxidant response to resistance training.

Thus, in 2018, the American group of Lydia K. et al. [24] replicated this article with the same active subjects (19; 10 women and 9 men) with the same groups (HD vs. HL vs. PL), dose (HD 960 mg/day; LD 160 mg/day), duration of treatment (14 days), type of ginseng (KG) and physical exercise to be performed (five sets of leg press at 70%RM). In this research, the authors introduced qualitative and quantitative measures of ballistic jumps and questionnaires on muscle soreness and subjective perception of exertion previously used by Flanagan et al. [23], although they did not perform blood tests to check physiological markers. The results obtained from this intervention were a better response in the subjective perception of effort and a greater peak power in ballistic jumps in HD compared to LD and PL. On the other hand, the consumption of KG, regardless of the dose, improves the perception of muscle discomfort after strength exercise compared to PL. The authors concluded that the subjective perception of the participants regarding effort and muscle pain was attenuated by the consumption of KG. Furthermore, the consumption of high doses of KG demonstrated a lower neuromuscular fatigue response after strength training.

Jifu Wu et al. [25] carried out a double-blind intervention with randomised crossover groups with intake of Rg1 (5 mg), the main component of ginseng, with 12 healthy men. This intervention consisted of intake 1 h before performing an incremental test on a cycle ergometer for 1 h at 70% VO2max. Subjects underwent a biopsy before, immediately after, and 3 h after exercise. In this study, the results reflected that the experimental group attenuated the glutathione response, as well as improved satellite cell replenishment and transient myogenic induction response. These results led this research group to the conclusion that Rg1 consumption improves the renewal and regeneration of muscle fibers after physical exercise. This article is a reference regarding the fibrillar processes that Rg1 consumption can cause, and it is the only article found that performs muscle biopsies during the intervention, so its results and conclusions shed clarity on the biological processes underlying the results of other studies with other types of variables evaluated.

Hyun Lyun Jung et al. [26], in 2020, carried out a randomised clinical trial with crossover groups evaluating the intake of WGE at doses of 700 mg/day for 5 days in 10 healthy subjects (men). In this intervention, the exercise performed was 20 min of continuous downward treadmill running (−10% incline at 60% VO2max) to emphasise eccentric contraction, followed by five sets of drop jumps from 60 cm. After completing the exercise, WGE was administered immediately after and for the following 4 days. On days 1, 3 and 5, a cognitive test was carried out by computer, as well as a physical test, perception of muscle damage and a blood test (cortisol, IL-6, myoglobin and antioxidant capacity). The results of this clinical trial did not reveal any differences between groups. Therefore, the conclusions carried out by these authors revealed that WGE intake has no effect on the biological processes produced by eccentric exercise.

A year later, Yi-Ming Chen et al. [27] exposed 20 healthy women to a randomised parallel-group trial comparing the intake of 2 g/day of JG for 6 weeks. In this study, the subjects carried out a day of familiarisation prior to the tests to be carried out, and later, a full day of tests was carried out, in which blood extraction, anthropometric measurements, jump test for kinematic and kinetic analysis, test on a cycle ergometer and perception of fatigue were performed. In the incremental cycle ergometer test, the exhaustion time performed by each subject was evaluated, in addition to serving to induce fatigue. The results revealed significant differences between both groups. Regarding the biochemical analysis, hepatic markers (SAT and ALT) decreased in the experimental group, as did CK levels; on the other hand, creatine and one of the lipid parameters, such as HDL, increased in the group that consumed JG. Also, in this section, there was an increase in glucose and FFA levels in the experimental group. Regarding the anthropometric components, fat mass decreased in the JG group. Results derived from the kinetic drop jump test revealed that the GS group experienced an increase in interaction reactive strength in 70 DJs. Regarding the kinematic aspect, the stiffness of the right ankle and the stiffness of the knee decreased by 40 DJs and 30 DJR and 50 DJL, respectively. Finally, regarding exhaustion time, the GS group increased time and VO2max. These authors summarised the results of their article on the improvement regarding the decrease of fatigue-related markers after exercise and improvement in glucose levels after strenuous exercise. In this way, the conclusions of this article also provide evidence regarding the possibility of reducing the risk of musculoskeletal injury due to the consumption of JG in healthy women.

Gislaine Cristina-Souza et al. [28] reported a better response in muscle excitation and subjective perception of effort due to the consumption of 100 mg/kg/day of PG in 10 athletic subjects after eccentric strength work. This is the only randomised intervention study with crossover groups in which the subjects analyzed are athletes. The study was carried out for 8 days with a separation of 7 days to avoid effects from one treatment to another. Subjects began ingesting PG or PL with individually agreed doses. On the fifth day, they performed a test that consisted of four sets of squats with muscle tension at 70% RM until concentric failure, with special emphasis on the eccentric phase. The next 3 days were for observation and evaluation. In this intervention, the electrical activity of the vastus lateralis muscle (EMG), the subjective perception of effort and physiological markers were evaluated through blood extraction. These evaluations were carried out before, after and 24, 48 and 72 h after exercise. The conclusions drawn from this study assume that PG consumption improves both muscle excitability and the perception of effort without significant differences in the rest of the parameters.

Two of the most recent studies are Ching-Hung Lin et al. [29] and Yi Yang et al. [30]. The first of the studies carried out a clinical trial in a context similar to the first analyzed study by Cheng-Chen Hsu [19]. In this field, the authors of the recent study evaluated 14 active male subjects organised randomly and in crossover groups, comparing the intake of AG in the same doses (1.6 g/day) during a similar period (30 days) with the PL group. In this case, the physical activity to be evaluated was a treadmill run with a downward incline. The evaluations to be carried out were done through blood extraction and the pain perception questionnaire. The results are similar to the first study carried out in this area, with an improvement in the CK response and, in addition, a decrease in lipid peroxidation. In conclusion, these authors ratify the conclusions of the article written 16 years earlier, in which an improvement in the reduction of muscle damage induced by exercise is evident. Yi Yang et al. [30] carried out a randomised intervention in two parallel groups with 110 healthy subjects, between men and women. In this case, the intake of 420 mg/day of KG for 8 weeks was evaluated. The exercise that was performed to induce physiological processes was an incremental test on a cycle ergometer. Like most of the articles reviewed, blood extraction and subjective perception, in this case of physical strength, were the variables studied. The results collected by the study reflected an improvement in CK and lactate levels in the group that ingested KG. At the same time, regarding the subjective scale of physical strength, the results were also favorable towards the experimental group. In conclusion, these authors defined that KG consumption improves muscle damage, neuromuscular fatigue and subjective indicators regarding strength.

## 4. Discussion

### 4.1. Effect of Ginseng on Muscle Damage

Of the 13 studies reviewed, 6 of them show positive results on ginseng supplementation and physiological markers of muscle damage. The type of ginseng used for these six studies were AG (*n* = 2), KG (*n* = 2), JG (*n* = 1) and RG (*n* = 1). Of these, three of them evaluated treadmill running (ascending, descending and normal plane), two evaluated cycle ergometer work and one evaluated strength work. Due to the variety of exercises to be performed, we can see how the response to muscle damage occurs in exercises that focus on both concentric and eccentric contraction work.

The procedure used to assess biomarkers of muscle damage was through blood extraction. The main studied markers for muscle damage are CK and IL-6. Of the eight studies that used blood extraction to compare these markers, only two did not find changes in these parameters. The ginseng variants used in these two studies were PNG [21] and WGE [26]. The doses established in the articles varied from 420 mg/day to 2 g/day. In the article in which the ingestion of PNG occurs, the subjects used were athletes, and the intervention was specific on a single day and a single intake prior to performing the exercise. The physiological efficiency of these athletes, together with a lower overall intake than the rest of the articles studied, makes more research necessary on the influence of chronic intake on this type of subject. In the article that evaluated the intake of WGE and showed no differences in muscle damage, the intervention was again acute, although the follow-up was carried out for 5 days in total. The articles that evaluated where significant differences were found in the intake period were carried out for at least 7 days and up to a maximum of 8 weeks. The differences in the chronic intakes in the rest of the article may cause the difference in the results on muscle damage.

In summary, the studies analyzed in this review suggest that a systematic and prolonged intake of ginseng can attenuate the response of CK and IL-6 as biological markers of exercise-induced muscle damage and inflammation. It is necessary to carry out more studies on this, as determining the typology of subjects, the type of ginseng, the intake dose, the intake period, and the exercise to be performed more specifically would facilitate learning the specific effects of this supplementation regarding muscle damage.

### 4.2. Effect of Ginseng on Muscle Fatigue

Muscle fatigue derived from exercise has been found to obtain significant differences in 4 of the 13 articles reviewed. In these articles, the intake of ginseng supplements KG (*n* = 2), RG (*n* = 1) and JG (*n* = 1) obtained significant benefits in healthy adult subjects. Regarding the intake of supplements, the amount ingested was variable, from 420 mg/day to an acute intake of 5 g/day. The duration of these protocols extended from an acute intervention study, 1 day, to the rest of the studies that assessed the effect in the medium-long term, carrying out studies for 2, 6 and 8 weeks. The exercise performed to induce said muscle fatigue was a cycle ergometer test in three of them and a strength test in the other. The most widely used exercise in these studies, whose results are significant with respect to muscle fatigue, is the incremental cycle ergometer test, a validated test to monitor the different effects of muscle fatigue [31]. In the article in which resistance exercise was used to induce muscular fatigue, this fatigue was measured through the deficit in the results of jumps, verifying the difference between groups regarding the loss of height. This tool is a fast, simple and non-invasive way to detect quantitative variations in fatigue levels [32]. Along with the execution of jumps, the most widely used fatigue measurer in the evaluation of high-intensity exercises is the level of accumulated blood lactate [33]. These ways of assessing fatigue have been used in the reviewed articles, giving valid support to the conclusions made by the authors.

After the results derived from the studies reviewed, we can conclude that the intake of ginseng has a direct implication on muscle fatigue, due in large part to the dampening of the appearance of lactate in the blood and the lower deficit experienced after performing jumps after exercise.

### 4.3. Effect of Ginseng on Muscle Regeneration

Muscle regeneration after exercise has an important role in the subject’s ability to sustain loads continuously [34]. As we have previously confirmed, the use of ginseng can improve muscle damage caused by exercise, but in addition to reducing muscle damage, can ginseng help with regeneration, making it more efficient for the body? Four of the reviewed studies reached an affirmative conclusion regarding this issue, using different types of ginseng in each of them—PG, Rg1, KG and JG. As in the entire review, the doses ingested by the participants were very varied, thus allowing us to determine a rigorous protocol for ginseng intake. In two of the articles, we can find a direct consequence between the decrease in muscle damage and the better muscle regeneration produced by the organism [23,27]. However, there are two articles that give greater support to this hypothesis due to the variables used and the results obtained. Firstly, Wu J. [25] and his group were the only ones who studied the effect of ginseng invasively, using muscle biopsies. In this way, they were able to observe how the intake of ginseng, Rg1 in this case, improved the replacement of satellite cells after exercise compared to those who did not ingest the supplement. Satellite cells are markers directly related to the body’s muscle regeneration [35]. Added to this discovery is the improvement of the transient myogenic induction response. Therefore, we obtain objective evidence on the direct relationship between ginseng consumption and muscle regeneration. On the other hand, Cristina-Souza [28] and her collaborators studied muscle regeneration of subjects through voluntary muscle excitation. In this article, it was found that those who used the PG supplement recovered muscle arousal levels after exercise more quickly than those who did not ingest it. In this way, we can observe in the analysis of the article that this regeneration is not directly related to less muscle damage because CK levels were not altered compared to those who did not take it. The variation in the protocols, as in the previous sections, is also diverse; these run between 1, 8, and 14 days and 6 weeks.

This section is especially important due to two of the articles that separate regeneration from muscle damage, processes that can become completely dependent. The intake of ginseng can improve both processes, emphasising each of them and providing muscle regeneration, an important way of improving the ability of subjects to adapt to continuous training stimuli.

### 4.4. Effect of Ginseng on Performance/Prevention

Exercising has positive effects on health [36]. However, the acute fatigue generated by exercise can increase the injury risk profile [37]. These arguments have led some of the studies reviewed to investigate the use of ginseng and its relationship with performance/prevention after exercising. Only three of the investigations used in this review have related these variables. Different types of ginseng—PG, KG and JG—were used in these articles. The intake of the food supplement varied depending on the chosen protocol, so the homogeneity of some results cannot be related to a specific intake of ginseng. Regarding the exercise performed, two of the investigated groups performed resistance exercise ([24,28]), while the other experimental group performed an incremental test on a cycle ergometer [27]. The two studies that used resistance exercise as a test to induce fatigue were performed with athletes and active adults, unlike the other study that used healthy subjects. The duration of these studies varied from 7 days to 6 weeks, the latter being where the most significant results regarding injury prevention were obtained. The evaluation of fatigue and its subsequent relationship with performance or risk of injury was carried out through voluntary muscle activation and jumping to verify the deficit between groups after resistance exercise. These two articles directly related fatigue to a decrease in the performance of the subjects. The article by Chen Y et al. [27] went a step further, evaluating the jumps qualitatively and observing the stiffness of the different joints involved in both bimodal and unipodal jumping. The experimental group with ginseng intake obtained better values in the kinematics of the jumping movement compared to those who did not consume the supplement. Evidence has shown that there are changes in the kinematics of the joints when fatigue appears in the subjects studied [38]. For this reason, the authors conclude that a decrease in fatigue from ginseng intake may help reduce the risk of injury, thus increasing athletic performance.

Physical activity contributes to maintaining and improving the health of the subjects who practice it [39]. In the same way, inactivity causes a deterioration in the development of the organism, which is harmful to health [40]. Injuries cause inactivity in subjects who exercise regularly, so prevention is one of the factors to take into account for an improvement in both sports and recreational performance and, of course, health. The reviewed articles that pay special attention to this aspect provide scientific support for the benefits of ingesting ginseng as a supplement that improves prevention and, consequently, physical exercise performance in those who consume it.

### 4.5. Effect of Ginseng on Subjective Perception

The perception of effort is used as a subjective quantifier of the individual perception of the physical demands of an activity [41]. That is why some articles use this tool as an evaluation of the body’s response to physical activity. In this review, three articles found significant results regarding the subjective perception of effort (RPE). The exercises performed by the studied groups were strength exercises [24,28] and incremental testing on a cycle ergometer [30]. The ginseng variants used in these investigations were PG and KG. The duration of intake with ginseng supplementation was 8 and 14 days and 8 weeks. The evaluation of RPE was carried out through questionnaires carried out immediately after the exercise, comparing the results between the experimental group and the control group. The three articles that used this variable identified significant differences between the control group and the experimental group. Thus, the authors concluded that the use of ginseng intake decreases the subjective perception of effort.

Questionnaires on the subjective perception of effort have been used as a tool to measure the physical demand induced by exercise since 1982 [42]. This type of questionnaire provides an easy and quick way of obtaining subjective information based on the internal load of the subjects. In this way, an improvement in individual perception is correlated with a better tolerance to efforts. Thus, the intake of ginseng can improve the ability to withstand efforts, as well as recover from them to maintain training programs or sports competitions, with little rest time between stimuli.

### 4.6. Limitations

One of the limitations of this review is the lack of homogeneity in the sports experience of the subjects studied in each of the articles reviewed. This aspect can cause the effect of the supplement used to generate different responses depending on the degree of sports preparation to which the subject is accustomed. On the other hand, the lack of homogeneity in the ginseng extracts used, as well as in the doses applied, can generate confusion about the greater or lesser influence of the supplement in the conditions studied. Finally, a significant limitation has been observed in the duration of the observed protocols, leading to the inability to answer a possible question about the acute or chronic effects of the supplement used. Future studies should take these aspects into account to draw more exhaustive conclusions about the intervention analyzed.

### 4.7. Applicability and Future Research

The present review reveals the relationship between ginseng and muscle damage induced by physical exercise. It has been observed how the consumption of this herbal supplement reduces aspects related to catabolism, such as muscle damage and fatigue, and improves the recovery process. There is scientific evidence that provides solid arguments about the belief of Traditional Chinese Medicine for the improvement of physical and mental energy, reducing fatigue because of less muscle damage. This supplement is already present in some of the best-known sports drinks in the world, but the use of ginseng in isolation can provide benefits for the general population and, more specifically, for athletes.

### 4.8. Future Research

The analysis of the benefits extracted from each of the studies, together with the limitations observed in the review, provides us with possible directions to address future studies on ginseng and its effects on exercise. It is necessary to provide more robust studies in terms of the methodology and the groups recruited to carry them out.

Therefore, the use of a certain type of ginseng should be an aspect to take into account to clarify the cause of the possible beneficial effects. Likewise, the use of experimental groups with experience in physical activity, and, more specifically, athletes, can increase the evidence on the use of this type of supplement, eliminating possible interference due to the status of the subjects studied. Finally, in future studies, the doses and duration of ginseng supplementation should take special importance. Determining the ideal dose without adverse effects, which maximises the possible benefits, should be an important focus to be considered by the authors. Likewise, know the time necessary for the improvements associated with ginseng to have the greatest efficiency and determine the benefits in the short, medium and long term. After this review, the belief in the importance of delving into this topic increases so that those who practice sports regularly and athletes can improve their health and performance in the regular periodisation of training.

## 5. Conclusions

Based on the documents consulted, possible positive effects of ginseng on muscle damage derived from physical exercise in healthy adults have been found when the intake of this supplement occurs compared to those who did not take it. Thus, it has direct consequences on fatigue, muscle regeneration, perception of effort, performance and injury prevention post-exercise. It is necessary to clarify the methodology to be carried out in each type of situation to provide more information on the improvement of ginseng supplementation.

## Figures and Tables

**Figure 1 nutrients-16-00090-f001:**
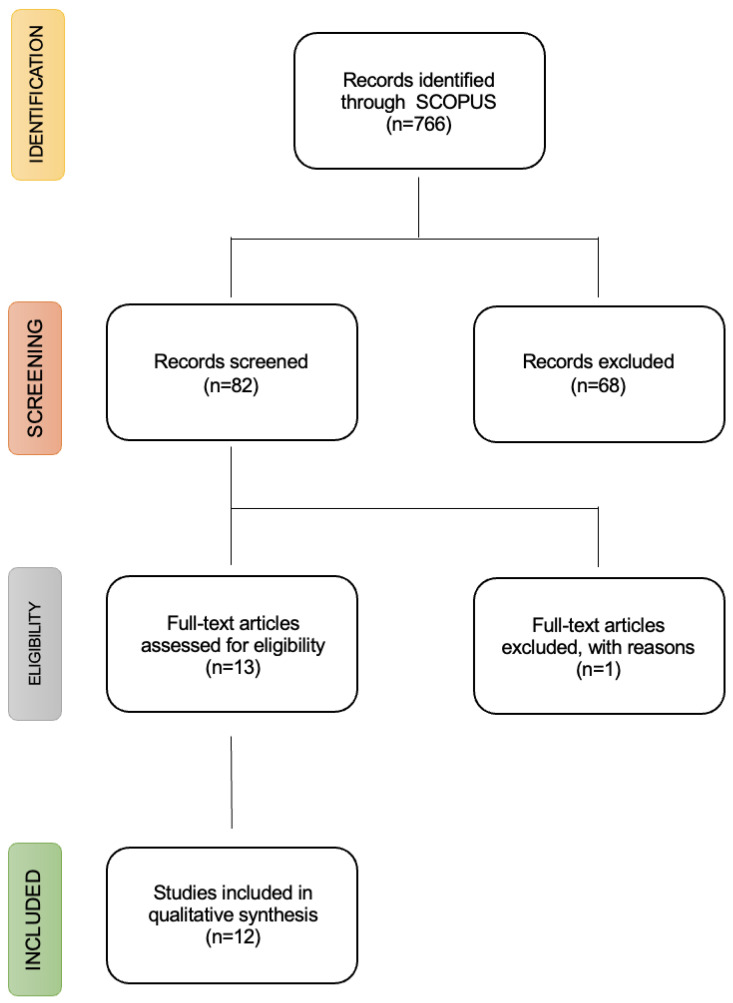
PRISMA flow diagram.

**Figure 2 nutrients-16-00090-f002:**
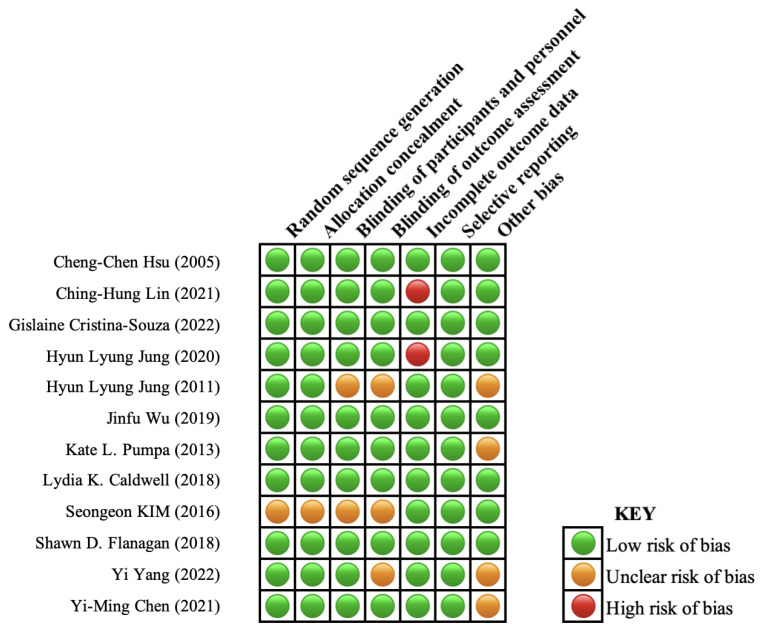
Risk of bias assessments by CCRBT [19,20,21,22,23,24,25,26,27,28,29,30].

**Table 1 nutrients-16-00090-t001:** Eligibility criteria.

**Study design**	Randomised and controlled clinical trials, cohort studies, cases and controls
**Population**	Healthy adults
**Intervention**	Ginseng intake as a dietary supplement
**Comparison**	Active adults without ginseng intake
**Outcomes**	Observation of beneficial physiological results on muscular structures in healthy adults after performing physical exercise

**Table 2 nutrients-16-00090-t002:** Article inclusion and exclusion criteria.

Effect of ginseng intake on muscle damage induced by exercise in healthy adults.				
**Eligibility**				
Screening according to information available in the title and/or abstract and study design				
What physiological benefits can healthy adults obtain from regular intake of ginseng compared to those who do not ingest this supplement?				
Can muscle damage induced by training be attenuated by the consumption of ginseng compared to its non-ingestion in healthy adults?				
Is there a positive relationship between ginseng intake in the recovery of muscle injuries in healthy adults compared to those who do not consume it?				
**Information**		**Yes**	**No**	**¿?**
It specifically talks about the intake of ginseng		X		
It considers any variety of ginseng		X		
It considers sporadic consumption of ginseng		X		
It considers some muscle damage not from performing exercise			X	
It considers subjects with some type of health problem			X	
The patients are not adults			X	
**Design**		**Yes**	**No**	**¿?**
Human studies		X		
Meta-analysis		X		
Systematic review		X		
Observational studies	Report and case series		X	
	Cross-sectional		X	
	Populations		X	
	Cases and controls	X		
	Cohorts	X		
Experimental trials	Randomised, blind	X		
	Randomised, double-blind	X		
	Not randomised		X	
Animal studies (laboratory)			X	
In vitro studies			X	
The summary will be ACCEPTED if:				
	There is a YES in the information section and a If we doubt a priori (?) in the design,			
	Or if we doubt (?) the information but the design has a YES or ¿?			
	Or if there is only one title (No Summary appears) and we do not know if it is of interest or not (?)			
In all other cases, the Abstract will NOT be ACCEPTED in the screening.				

**Table 3 nutrients-16-00090-t003:** Data extraction protocol WG3-A8.

**Aim of study**	The aim of the review was to determine whether ginseng intake may help with the ability to recover from muscle damage induced by exercise in healthy adults.
**Health area**	Nutrition
**Study design**	Intervention
**Articles/reviews analysed**	12 studies, number of SR NA
**Assessed steps**	·All steps of a systematic review
	·Formulate review question
	·Write the protocol
	·Device search strategy
	·Search
	·De-duplicate
	·Title-abstract screening
	·Obtain full-text
	·Full-text screening
	·Data extracting
	·Critical appraisal
	·Synthesise data
	·Write up review
	·Translation

**Table 4 nutrients-16-00090-t004:** Main data from the reviewed articles.

Author, Year, Country	Study Design	Type, Doses, Duration	Compared Groups	Performance Test	Outcome Measures	Authors’ Conclusions
Cheng-Chen Hsu et al., (2005) Taiwan [19]	R, DB, CO	AG 1.6 g/day 4 weeks	GE vs. PLACEBO	Incremental running test (treadmill)	GE improves CK response	↓ Muscle damage
Ching-Hung Lin et al., (2021) Taiwan [29]	R, DB, CO	AG 1.6 g/day 30 days	GE vs. PLACEBO	Downhill running (treadmill)	GE improves CK response GE attenuates lipid peroxidation	↓ Muscle damage
Gislaine Cristina-Souza et al., (2022) Brazil [28]	R, DB, CO	PG 100 mg/kg/d 8 days	GE vs. PLACEBO	Eccentric strength	GE improves RPE response GE improves response in muscle excitement	↑ Muscular excitement ↓ RPE ↑ Muscle regeneration
Hyun Lyung Jung et al., (2020) Korea [26]	R, DB, CO	WGE 700 mg/day 5 days	GE vs. PLACEBO	Downhill running (treadmill)	There are no differences	–
Hyun Lyung Jung et al., (2011) Korea [20]	R, DB	RG 20 g/day 7 days	GE vs. PLACEBO	Uphill running (treadmill)	GE improves CK levels after 72 h GE improves IL-6 levels 2 and 3 h later GE decreases insulin at 90 min. GE reduces glucose levels in 60 min.	↓ Muscle damage ↓ Muscle inflammation
Jinfu Wu et al., (2019) Taiwan [25]	R, DB, CO	Rg1 5 mg 1 day	GE vs. PLACEBO	Incremental cycling test (Cycle ergometer)	GE attenuates glutathione response GE improves post-exercise satellite cell replacement GE enhances transient myogenic induction response	↑ Muscle regeneration
Kate L. Pumpa et al., (2013) Australia [21]	R, DB, P	PNG 1 g 1 day	GE vs. PLACEBO	Downhill running (treadmill)	There are no differences	–
Lydia K. Caldwell et al., (2018) EEUU [24]	R, DB, CO	KG HD 960 mg/day; LD 160 mg/day 14 days	HD vs. LD vs. PLACEBO	Strength test	HD improves response in RPE HD and LD improve perception of muscle discomfort HD improves peak power in ballistic jumps	HD vs. LD/PL: ↓ RPE ↓ Neuromuscular fatigueHD/LD vs. PL: ↓ Muscle pain
Seongeon KIM et al., (2016) Korea [22]	R, DB, CO	RG—5 g/day—1	GE vs. PLACEBO	Anaerobic power test (cycle ergometer)	GE improves lactate levels 30 and 60 min later GE improves ammonium levels after 30 min.	↓ Neuromuscular fatigue
Shawn D. Flanagan et al., (2018) EEUU [23]	R, DB, CB	KG HD 960 mg/day; LD 160 mg/day 14 days	HD vs. LD vs. PLACEBO	Strength test	HD suppresses cortisol response HD decreases, CK increases HD increases guttation in 30 and 60 min later HD increases SOD HD improves TAP	HD vs. LD/PL: ↓ Muscle catabolism ↑ Acute antioxidant response
Yi Yang et al., (2022) China [30]	R, DB, P	KG 420 mg/day 8 weeks	GE vs. PLACEBO	Incremental cycling test (cycle ergometer)	GE improves CK response GE improves lactate response GE improves the subjective perception of strength	↓ RPE ↓ Muscle damage ↓ Neuromuscular fatigue
Yi-Ming Chen et al., (2021) China [27]	R, DB, P	JG 2 g/day 6 weeks	GE vs. PLACEBO	Incremental cycling test (cycle ergometer)	GE reduces indicators of liver damage GE improves CK levels GE increases creatine level GE increases HDL level GE decreases fat mass GE increases exhaustion time GE improves glucose and FFA levels GE increases RSI by 70 DJ GE decreases right ankle stiffness in 40 DJs GE decreases knee stiffness in 30 DJR and 50 DJL	↓ Muscle damage ↓ Neuromuscular fatigue ↑ Muscle regeneration ↑ Glucose and FFA ↓ Injury risk

AG, American ginseng; BUN, blood urea nitrogen; CB, counterbalance; CK, creatine kinase; CO, crossover group; DB, double-blind; DJ, drop jump; DJL, drop jump left; DJR, drop jump right; FFA, free fatty acids; GE, experimental group; M, male; HD, high dose; HDL, high-density lipoprotein; IL-6, interleukin 6; JG, Jilin ginseng; KG, Korean ginseng; LD, low dose; F, female; P, parallel; PG, Panax ginseng; PNG, Panax notoginseng; R, random; RG, red ginseng; RPE, rate of perceived exertion; RSI, relative strength index; SOD, superoxide dismutase; TAP, total antioxidant performance; WGE, wild ginseng extract. Up arrow ↑ means “increase”; ↓: Down marrow means “decrease”.

## Data Availability

Not applicable.

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
