# Peer review of "Effect of Ginseng Intake on Muscle Damage Induced by Exercise in Healthy Adults"

_nutrients, 2023, doi:10.3390/nu16010090_

Round 1
Reviewer 1 Report
Comments and Suggestions for Authors
Thank you for the opportunity to review this MS. In general, this is a well-written narrative review. However, some parts of the MS require clarification. Please find my comments and suggestions below:
Line 119: Why only Scopus ? What about Pubmed and WOS ? Please elaborate on this.
Line 122: What was the time-related starting point for database search here ?
Figure 1.
Any info on the screening criteria ? What made the 766 drop to 82 papers ?
Table 4. Is there any chance you could expand on these findings and outline by how much muscle damage and inflammation markers improved ?
Line 239: VO2 should be presented as V̇O2, as this is the correct way to indicate flow, here and in the MS.
Line 215: By how much was this improvement in IL-6 concentration ?
Line 240-281: This looks more like a book chapter, rather than a scientific MS and its results section. The point of this is also unclear. Please consider re-writing this section of the MS. Same goes from line 300-391.
Line 290: Again, by how much did the cortisol level decrease ?
Line 571: In the conclusion section, I would love to see the magnitude of effect here. This would then help providing guidelines related to dosage and subsequent effects of Ginseng supplementation on muscle recovery.
Comments on the Quality of English Language
Please give this MS to an official proofreading service. Minor issues are detected.
Author Response
Dear reviewer in the attached document the answer. Thank you very much for your words and recommendations.

Reviewer 2 Report
Comments and Suggestions for Authors
Solid work by the authors.
Structured work with a solid theoretical introduction. From the very beginning, precise research tools.
Poor group size for 11 studies and too large number in one study, as many as 110.
There is not enough precise numerical information on inflammatory parameters such as keratin kinase or interleukins.
Author Response

(The authors gave the same response as above.)
